# Global implementation research capacity building to address cardiovascular disease: An assessment of efforts in eight countries

Mary Beth Weber[1]*, Ana A. Baumann[2], Ashlin Rakhra[3], Constantine Akwanalo[4], Kezia Gladys Amaning Adjei[5], Josephine Andesia[6], Kingsley Apusiga[5], Duc A. Ha[7], Mina C. Hosseinipour[8], Adamson S. Muula[9], Hoa L. Nguyen[10], LeShawndra N. Price[11], Manuel Ramirez-Zea[12], Annette L. Fitzpatrick[13], Meredith P. Fort[14]

1 Hubert Department of Global Health, Emory University, Rollins School of Public Health, Atlanta, Georgia, United States of America, 2 Division of Public Health Sciences, Department of Surgery, Washington University in St. Louis, St. Louis, Missouri, United States of America, 3 Department of Population Health, NYU Grossman School of Medicine, New York City, New York, United States of America, 4 Department of Medicine, Moi Teaching and Referral Hospital, Eldoret, Kenya, 5 Department of Physiology, School of Medicine and Dentistry, Kwame Nkrumah University of Science & Technology, Kumasi, Ghana, 6 Academic Model Providing Access to Healthcare (AMPATH), Eldoret, Kenya, 7 Health Strategy and Policy Institute, Vietnam Ministry of Health, Hanoi, Vietnam, 8 Department of Medicine, University of North Carolina at Chapel Hill, Chapel Hill, North Carolina, United States of America, 9 College of Medicine, University of Malawi and the Kamuzu University of Health Sciences, Blantyre, Malawi, 10 University of Massachusetts Chan Medical School, Boston, Massachusetts, United States of America, 11 National Institute of Allergy and Infectious Diseases, National Institutes of Health, Bethesda, Maryland, United States of America, 12 INCAP Research Center for the Prevention of Chronic Diseases, Institute of Nutrition of Central America and Panama (INCAP), Guatemala City, Guatemala, 13 School of Public Health, University of Washington, Seattle, Washington, United States of America, 14 Colorado School of Public Health, University of Colorado – Anschutz Medical Campus, Aurora, Colorado, United States of America

* mbweber@emory.edu

**Data Availability Statement:** All relevant data are within the paper and its Supporting information files.

## Abstract

Cardiovascular diseases are the leading causes of morbidity and mortality worldwide, but implementation of evidence-based interventions for risk factors such as hypertension is lacking, particularly in low and middle income countries (LMICs). Building implementation research capacity in LMICs is required to overcome this gap. Members of the Global Research on Implementation and Translation Science (GRIT) Consortium have been collaborating in recent years to establish a research and training infrastructure in dissemination and implementation to improve hypertension care. GRIT includes projects in Ghana, Guatemala, India, Kenya, Malawi, Nepal, Rwanda, and Vietnam. We collected data from each site on capacity building activities using the Potter and Brough (2004) model, mapping formal and informal activities to develop (a) structures, systems and roles, (b) staff and infrastructure, (c) skills, and (d) tools. We captured information about sites' needs assessments and metrics plus program adaptations due to the COVID-19 pandemic. All sites reported capacity building activities in each layer of the Capacity Pyramid, with the largest number of activities in the Skills and Tools categories, the more technical and easier to implement categories. All sites included formal and informal training to build Skills. All sites included a baseline needs assessment to guide capacity building activities or assess context and inform intervention design. Sites implementing evidence-based hypertension interventions

**Funding:** The research reported in this publication was supported by the National Heart, Lung, and Blood Institute (NHLBI) under award numbers HL136789, HL136790, HL136791, HL138631, HL138635, HL138638, HL138647, and HL151310. The content is solely the responsibility of the authors and does not necessarily represent the official view of the NHLBI, the National Institute of Allergy and Infectious Diseases, the National Institutes of Health or the U.S. Department of Health and Human Services. The Capacity Building Subcommittee included members from the funding organization who contributed to the work here, including data analysis, preparation of the manuscript, and decision to publish; however, the funders had no role in conduct and assessment of the individual projects included in the GRIT network.

**Competing interests:** The authors have declared that no competing interests exist.

used common implementation science frameworks to evaluate implementation outcomes. Although the COVID-19 pandemic affected timelines and in-person events, all projects were able to pivot and carry out planned activities. Although variability in the activities and methods used existed, GRIT programs used needs assessments to guide locally appropriate design and implementation of capacity building activities. COVID-19 related changes were necessary, but strong collaborations and relationships with health ministries were maintained. The GRIT Consortium is a model for planning capacity building in LMICs.

## Introduction

Cardiovascular diseases (CVDs) are leading causes of mortality and morbidity; the top two causes of death globally are ischemic heart disease followed by stroke, accounting for 16% and 11% of the total deaths, respectively [1]. Hypertension, a major driver of CVD, is the leading attributable risk factor for death worldwide [2]. Proven prevention and management programs for hypertension and other CVD risk factors exist [3–6], but translation of these models into real-world settings is needed. This is particularly true in low and middle income countries (LMICs), where the burden of chronic diseases is high [2, 7], but capacity for implementation research is limited, presenting a barrier for accelerating the implementation of evidence-based interventions [8–10].

To address these barriers and the gaps in implementation research and practice in LMICs [9], the National Heart, Lung and Blood Institute (NHLBI) of the United States (U.S) National Institutes of Health funded eight research grants under two funding opportunity announcements for partnering interdisciplinary teams of non-U.S. and U.S. research investigators: (1) Hypertension Outcomes for T4 Research in Lower Middle-Income Countries (Hy-TREC) and (2) T4 Translation Research Capacity Building Initiative in Low Income Countries (TREIN). The eight countries represented by these projects face challenges to implementation of prevention and control of non-communicable diseases (NCDs) such as hypertension and CVD that are seen across LMICs. LMICs have fewer health resources (including personnel and financial resources) to address the dual burden of chronic and infection diseases as well as continued needs for maternal and child health [11, 12]. Many LMICs (e.g., Kenya, India, Guatemala) organize their health systems in tiers with primary, secondary and tertiary care facilities; however referral networks between these care levels are inefficient in some settings, leading to gaps in care management (for examples, see data from Kenya [13] and India [14]).

Furthermore, the healthcare systems in LMICs face specific barriers to managing conditions like hypertension. Examples include poor or inconsistent access to medications in Ghana [15], Malawi, and Guatemala [16]; low rates of disease diagnosis in Malawi [17]; heterogeneity in treatment guidelines, diagnostic testing, and available medications leading to inconsistency in treatment for patients with hypertension in India [18–20]; insufficient access to guidelines for hypertension prevention and management in Rwanda [21]; and insufficient staffing and high staff turnover in Guatemala [16]. Healthcare providers, policy makers, and patients in some LMIC settings often rely on incorrect, missing, or limited knowledge of hypertension management in providing patient or self-care and interacting with the healthcare system (e.g., in Guatemala [16, 22] and Vietnam [23]).

Across LMICs, there is variability in national policies and healthcare resource allocation for hypertension and other NCDs. In Nepal, for example, there is a lack of national policies to address and health care system training and resources (human and financial) for prevention or management of hypertension and other NCDs [24]. On the other end of the spectrum, Ghana

has a national strategy to deliver basic community-based health services, which has resulted in increased access to care [25, 26]; however, restrictions on who can administer medications, service reimbursement issues, and lack of knowledge of and training on hypertension among community health workers, has reduced the impact on hypertension management [27]. Similarly, in Vietnam, significant and rapid economic growth since the 1980's has resulted in reductions in poverty, improvements in living standards, and high insurance coverage, but this growth has also accelerated the epidemic transition leading to a rapid increases in NCDs including hypertension which have outpace the growth in screening programs and patient self-management education [28–30]. Finally, country-specific data on disease rates and patterns, patient and economic burden, risk factors, and programs are lacking in many LMIC settings particularly for NCDs (for example, see studies from Rwanda [21], Nepal [24], and Malawi [31]), making it challenging to design and implement appropriate interventions.

Given the shared research interests of Hy-TREC and TREIN awardees in intervention development and implementation and building capacity for implementing late-stage translation research, the NHLBI convened the awardees as an investigator network, which became known as the Global Research on Implementation and Translation Science (GRIT) Consortium. The GRIT Consortium fosters collaborations to advance implementation science research and research capacity to reduce the global burden, particularly in LMICs, of hypertension and CVDs. To enable other researchers, public health practitioners, and clinicians to learn from the models created across the GRIT Consortium, there is a need to describe the capacity building efforts across the network.

Research capacity building can be defined as "an ongoing and iterative process of empowering individuals, interdisciplinary teams, networks, institutions and societies to identify health and health-related challenges; to develop, conduct and manage scientifically appropriate and rigorous research to address those challenges in a dynamic and sustainable manner; and to share, apply and mobilize research knowledge generated with the active participation of engaged stakeholders and decision-makers" [32]. Potter and Brough developed a model for systemic capacity building to operationalize activities to improve success (see Fig 1) [33]. They proposed a four-tier hierarchy of needs when building capacity in a system and programs should address capacity at each of these levels: 1. Structures, systems, and roles; 2. Staff and facilities; 3. Skills; and 4. Tools. These four levels are interdependent, each promoting success (or failure) of the preceding and proceeding levels. Within each tier, there is a need to improve specific capacities including performance (Tools), personnel (Skills), workload, supervisory, facility, support (Staff and Facilities), and systems, structure and role (Structures, Systems, and Roles) capacities. To date, few studies provide detailed reporting of capacity building activities, and the GRIT consortium is a unique opportunity to present a descriptive analysis of the types

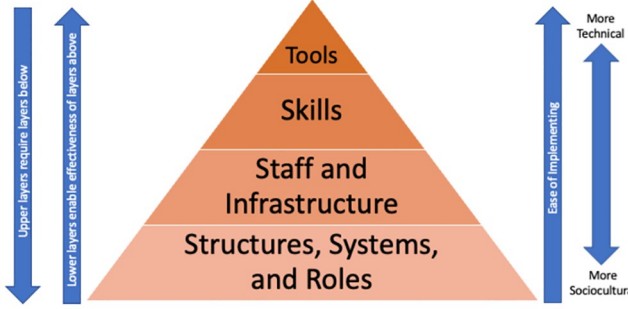

**Fig 1. Potter and Brough Capacity Pyramid.**

of capacity building activities across multiple LMIC settings. Characterization of capacity building activities in LMICs, using frameworks like the one developed by Potter and Brough, is an important contribution to the literature and for practitioners. These data provide key details on the types of activities being implemented in real-world capacity building efforts and offer considerations for researchers and implementers.

Herein, we describe the capacity building activities across the GRIT Consortium. We use the systemic capacity building model [33] to categorize and describe the capacity building activities by site. In addition, we report on the needs assessments and metrics across the GRIT projects as well as adjustments made to capacity building activities during the COVID-19 pandemic.

## Materials and methods

### The Global Research on Implementation and Translation Science (GRIT) consortium

In 2017, NHLBI funded five Hy-TREC projects that tested regional and national implementation strategies for evidence-based interventions for prevention, treatment, and control of hypertension and three TREIN projects focused on building transdisciplinary, in-country research capacity in CVD and dissemination and implementation sciences. A brief description of the projects' teams, location, and goals is shown in Table 1 (additional program/intervention details can be found in the project-specific methods publications [13, 14, 24, 28, 31, 34, 35]).

### Reporting and categorization of capacity-building activities

The Capacity Building Subcommittee of the GRIT Consortium, which includes representatives of each of the GRIT projects, GRIT coordinating center members, and NHLBI staff, aims to (1) develop plans for sharing training materials, design and implement training efforts and standardize implementation research training across the GRIT consortium; (2) assess the types of training necessary to build local and regional research capacity and develop training activities to build these skills; (3) identify opportunities to leverage in-country investments to build capacity; and (4) develop metrics of training programs to evaluate trainee and program success. For the analysis reported here, Capacity Building Subcommittee members created a data table (S1 Appendix) to collect data on capacity building activities and metrics as well as plans for project needs assessments based on the systemic capacity building model [33]. The data collection tool includes items for individual sites to report project-related capacity building purchases and activities in the following categories/subcategories:

- Tools: Equipment, Computers/Digital Devices/Cell Phones, Internet/Cell Phone Access/ Implementation Science Frameworks, Other

- Skills: Formal Training (planned and structured training including courses, lectures, training classes), Informal Training (unplanned/unstructured training including mentoring activities), Curriculum Used, Other

- Staff and Infrastructure: Mentors, Investigator Time (protected), Funding, Other

- Structures, Systems, Roles: New Positions Developed, Ministry of Health Involvement, Policy Development, Forums/Dissemination/etc., Other

- Metrics: Implementation Outcomes, Clinical Outcomes, Other

- Needs Assessments Plans

**Table 1. Global Research on Implementation and Translation Science (GRIT) consortium member projects.**

| Country Type | Project Title | LMIC Partner | U.S./Other HIC Partner(s) | Primary Objective |
|---|---|---|---|---|
| Ghana Hy-TREC | Uptake of task-shifting strategy for blood pressure control in community health planning services: a mixed methods study [34] | Kwame Nkrumah University of Science and Technology; Kintampo Health Research Centre | Saint Louis University; NYU School of Medicine; NYU Langone Health | To improve uptake and evaluate in a mixed methods study the role of practice facilitation and task-strengthening strategies for hypertension control (TASSH) at 70 Community-based Health Planning and Services Compounds. |
| Guatemala Hy-TREC | Implementing a multicomponent intervention to improve hypertension control in Central America [35] | Institute of Nutrition of Central America and Panama (INCAP), Guatemala; Ministry of Health and Social Welfare of Guatemala; Center of Excellence for Cardiovascular Health (CESCAS), Argentina; Instituto de Efectividad Clinica y Sanitaria | Tulane University; University of Colorado-Denver | To plan, implement and evaluate a multilevel and multicomponent hypertension control program for patients with uncontrolled hypertension in the primary healthcare system in rural Guatemala. |
| India Hy-TREC | Integrated tracking, referral, and electronic decision support, and care coordination (I-TREC) [14] | Centre for Chronic Disease Control; All India Institute of Medical Sciences; Public Health Foundation of India | Emory University | To assess the effectiveness of the Integrated Tracking, Referral, Electronic Decision Support and Care Coordination Package (I-TREC) package for improving hypertension and diabetes processes of care in patients at community health clinics in Punjab State. |
| Kenya Hy-TREC | Strengthening Referral Networks for Management of Hypertension Across the Health System (STRENGTHS)[13] | Moi Teaching and Referral Hospital; Moi University College of Health Sciences; Academic Model Providing Access to Healthcare (AMPATH) | NYU School of Medicine; Purdue University; Duke University; University of Southern California | To utilize transdisciplinary implementation research strategies to address the challenge of improving hypertension control in Western Kenya and measure the program's effectiveness and cost-effectiveness. |
| Vietnam Hy-TREC | Conquering Hypertension in Vietnam: Solutions at Grassroots level [28] | Health Strategy and Policy institute, Ministry of Health, Vietnam | University of Massachusetts Medical School | To evaluate the implementation and effectiveness of two multi-faceted community and clinic-based strategies on the control of elevated blood pressure among adult men and women. |
| Malawi TREIN | NCD-BRITE (Building Research Capacity, Implementation, and Translation Expertise) [31] | The University of Malawi College of Medicine; Ministry of Health and Population, Malawi; Dignitas International; Partners in Health, Malawi | University of North Carolina Chapel Hill; London School of Hygiene and Tropical Medicine | To build capacity for sustainable late-stage T4 translation research for heart, lung, blood and sleeping diseases and disorders in Malawi and design a context specific research plan for implementing T4 translation research for these conditions. |
| Nepal TREIN | Translational Research Capacity Building Initiative to Address Cardiovascular Diseases in Nepal [24] | Dhulikhel Hospital Kathmandu University Hospital; Kathmandu Medical College; Nepal Health Research Council; B.P. Koirala Institute of Health Sciences | University of Washington; Yale University | To build national capacity in Nepal to lead transnational research in cardiovascular disease by creating and training a multi-sectorial, multidisciplinary team; systematically assessing national needs; and developing an actionable translational research plan for the prevention and management of CVD. |
| Rwanda TREIN | Developing T4 translational research capacity for control of hypertension in Rwanda | Regional Alliance for Sustainable Development (RASD Rwanda); Ministry of Education Rwanda; Ministry of Health Rwanda; Gent University; King Faisal Hospital Kigali; University Teaching Hospital of Kigali; University of Rwanda | Washington University in Saint Louis | To increase uptake of proven hypertension control strategies by building competencies in T4 translation research and creating a collaborative team of healthcare providers, researchers, and public health experts. |

Key: LMIC = low to middle-income country; HIC = high income country; Hy-TREC = Hypertension Outcomes for T4 Research in Lower Middle-Income Countries; TREIN = T4 Translation Research Capacity Building Initiative in Low Middle-Income Countries

Data collection instruments were emailed to each GRIT program Principal Investigator (PI) in Summer 2019, and a team member was tasked with providing the requested information. Completed surveys were returned to the Capacity Building Subcommittee for review and analysis. In Spring and Summer of 2021, the data instrument was adapted to allow for reporting of changes to capacity building activities due to the COVID-19 pandemic and data collection procedures were repeated.

Members of the manuscript writing team (MBW, AAB, AR, AF, MPF) conducted a thematic review of the information provided in the data collection tool. Each of the five reviewers reviewed for one category of data or the needs assessment; data were not double coded. Each reviewer pulled the data pertaining to their assigned category from the data collection tool, documented commonalities and differences across country sites, then created summary documents and tables including both quantitative (e.g., number of sites reporting various items) and qualitative (e.g., descriptions of needs assessments used) outcomes for each of the four tiers of the systemic capacity building model as well as a summary of each project's needs assessment activities, program/study metrics, and COVID-19 adaptations. Stakeholder checking was conducted by sending tables and supporting text to the sites' PIs and members of the Capacity Building Subcommittee who added additional detail and clarification. Adaptations to the capacity building activities due to COVID-19 were similarly reviewed and summarized. Additional updates were added upon review of the manuscript by members of the Capacity Building Subcommittee to reflect additional changes during the COVID-19 pandemic in April 2022 not captured by the updated data collection tool.

### Ethics approval

This descriptive study evaluates program-level data and is not human research; review board approval was not needed. The analysis reported in this manuscript did not include any patients or the public, collecting data only from the project teams through the PIs.

## Results

All eight GRIT projects reported capacity building activities (Table 2). Results are presented for each Tier of the model, starting with the easier to implement activities (those that are more technical, e.g., acquisition of tools)) at the top of the Capacity Pyramid through the harder to implement activities (those that require systemic or culture changes such as changes to structures and roles) at the bottom tier of the pyramid, followed by Metrics and Needs Assessments.

### Tools

Projects reported purchasing a variety of study equipment with computers, software (for data collection, analysis, or communications), and cell phones/SIM cards being purchased most frequently and across TREIN and Hy-TREC sites. Additional equipment purchases at individual sites included Wi-Fi Routers, printers, digital recorders, cameras, treadmills, educational materials, and internet access. Hy-TREC sites, reported purchasing medical equipment (e.g., blood pressure monitors, scales, stadiometers, measuring tapes, and glucometers).

Four Hy-TREC projects reported the use of an implementation science framework to assess program outcomes. These included Predisposing, Reinforcing, and Enabling Constructs in Educational/Environmental Diagnosis and Evaluation-Policy, Regulatory, and Organizational Constructs in Educational and Environmental Development (PRECEDE-PROCEED, Kenya), Reach, Effectiveness, Adoption, Implementation, and Maintenance (RE-AIM, India, Ghana, Guatemala), and Consolidated Framework for Implementation Research (CFIR, Ghana) [36–38].

**Table 2. GRIT consortium capacity building activities by domain and program type and site.**

| Capacity Domain | Activity/Components/ Deliverables | Program Site by Type | | | | | | | |
|---|---|---|---|---|---|---|---|---|---|
| | | Hy-TREC Sites Testing evidence-based intervention through implementation research | | | | | TREIN Sites Building research capacity for NCD and D&I research | | |
| | | Ghana | Guatemala | India | Kenya | Vietnam | Malawi | Nepal | Rwanda |
| Tools | Medical Equipment | ✓ | ✓ | ✓ | ✓ | ✓ | ✓ | - | - |
| | Computer Equipment (computers, tablets), Cell Phones, or Software | ✓ | ✓ | ✓ | ✓ | ✓ | ✓ | ✓ | - |
| | Internet/Cell phone access | - | ✓ | ✓ | ✓ | ✓ | ✓ | ✓ | - |
| | IS Framework (if used) | ✓ | ✓ | ✓ | ✓ | - | - | - | ✓ |
| Skills | Formal Training | ✓ | ✓ | ✓ | - | ✓ | ✓ | ✓ | ✓ |
| | Informal Training | ✓ | ✓ | ✓ | ✓ | ✓ | ✓ | ✓ | ✓ |
| | Curriculum Used | ✓ | ✓ | ✓ | ✓ | ✓ | ✓ | ✓ | ✓ |
| Staff and Infrastructure | Mentors | - | ✓ | ✓ | ✓ | ✓ | - | ✓ | ✓ |
| | Investigator Time (protected) | ✓ | ✓ | ✓ | ✓ | ✓ | ✓ | ✓ | - |
| | Funding | ✓ | ✓ | ✓ | ✓ | ✓ | ✓ | ✓ | ✓ |
| | Other | - | - | - | - | ✓ | - | ✓ | - |
| Structures, System, and Roles | Involvement of Ministry of Heath | ✓ | ✓ | ✓ | ✓ | ✓ | ✓ | ✓ | ✓ |
| | Policy Development | - | ✓ | ✓ | ✓ | - | ✓ | - | ✓ |
| | Forums, Dissemination, etc. | ✓ | ✓ | ✓ | ✓ | ✓ | ✓ | ✓ | ✓ |
| Metrics | IS Process Outcomes | ✓ | ✓ | ✓ | ✓ | ✓ | ✓ | ✓ | ✓ |
| | Clinical Outcomes | ✓ | ✓ | ✓ | ✓ | ✓ | ✓ | - | - |

Key: Hy-TREC = Hypertension Outcomes for T4 Research in Lower Middle-Income Countries; TREIN = T4 Translation Research Capacity Building Initiative in Low Middle-Income Countries; NCD = Non-Communicable Disease; D&I = Dissemination and Implementation; IS = Implementation Science

## Skills

All projects reported some training activities, both formal and informal, to build skills among the project team and/or affiliated clinic staff. Formal training on hypertension was most commonly reported and was conducted at 6/8 GRIT sites. Guatemala provided the most comprehensive hypertension training program, a certification program for healthcare providers supported by the National Department of Training (DECAP—Departamento de Capacitación) and co-created with the Unit for Promotion and Education for Health (PROEDUSA) of the Ministry of Health. Vietnam invited experts from the National Heart Institute to conduct annual in-person training for study site doctors, nurses, and community health workers on hypertension diagnosis and management; Rwanda conducted a similar workshop to train healthcare providers. Four projects (India, Malawi, Guatemala, and Vietnam) conducted formal training in evaluation/data collection methods and research ethics for members of the project or study team and/or members of the Ministry of Health who were involved in implementing the project. Three sites (Guatemala, Rwanda, and Nepal) conducted formal training on dissemination and implementation research or other public health research topics. For example, Nepal delivered workshops on CVD research, qualitative research methods, and biostatistics. Rwanda conducted a weeklong training that covered hypertension, CVDs, and HIV, biostatistics, and dissemination and implementation research [21]. Malawi co-supported two PhD Fellows and provided formal manuscript, proposal, and protocol writing workshops.

Curricula for formal trainings varied with some sites using newly created materials and other sites leveraging existing programs (e.g., materials created by national or local

governments, the Global Alliance for Chronic Diseases, the University of Washington's Implementation Sciences Program, and the CITI training on Human Subjects Research). As reported previously [39, 40], all sites were invited by the Capacity Building Subcommittee and enrolled participants in a Massive Open Online Course (MOOC) on dissemination and implementation research developed and implemented by the Special Programme for Research and Training in Tropical Diseases at the World Health Organization [39].

All of the sites reported informal training activities on technology, engaging collaborators, and research skills. This training was delivered in person, by phone, or using communication software (e.g. Zoom). Some informal training leveraged other training and networking opportunities; for example, two sites (Guatemala, Nepal) facilitated the attendance of early stage investigators at international meetings and conferences.

### Staff and infrastructure

All sites reported capacity building activities to improve the skills, knowledge, and training of early stage investigators, researchers, clinicians or staff. Six sites reported planned mentoring of early stage team members by more experienced investigators. Mentors included the site PIs and Co-Investigators, including both in-country and U.S. experts. All sites except Rwanda used grant funds to provide protected time for investigators to ensure sufficient time and effort could be spent on the project. Two sites (Nepal and Vietnam) earmarked grant funding to pay support staff to assist with projects. Across sites, projects initiated new and strengthened existing partnerships and collaborations with partners within country, regionally (e.g., South-South collaborations between Guatemala and Argentina), and in High Income Country (HIC).

### Structures, systems, and roles

Although none of the project teams documented the creation of new, permanent positions as part of their project, some supported trainees or leveraged existing staff to assist in capacity building activities. For example, the Nepal team created sixteen research fellow positions as part of their project. All study teams reported collaborating with the Ministry of Health at the Central (and in some cases Regional) level as well as collaborations with local agencies (e.g., universities). Most sites included a Ministry of Health representative on the project's Steering Committee. In all of the Hy-TREC sites, Ministry of Health employees are implementing the intervention. The Multiple-PI of the Vietnam project is the Director of the Health Strategy, Policy Institute, a division of Vietnam's Ministry of Health.

Several sites described activities related to dissemination or policy. Project teams described meetings with key stakeholders including public health officials, community members, and community health workers to guide the projects and inform stakeholders of project/study design and progress. All sites described plans to disseminate results through local media, publications, and forums/symposia. Both Kenya and Guatemala have been involved in the development and/or dissemination of treatment guidelines.

### Metrics

All of the sites were collecting implementation outcome metrics, although these varied by site. Implementation outcome metrics included measures of training and capacity building ranging from number of trainees completing the program and achievement of deliverables by research fellows to evaluations of implementation climate, leadership support, and organizational capacity. Several sites included assessments of program adoption and sustainability. Five sites described multi-factor implementation evaluations. Guatemala and India are both assessing Reach, Effectiveness, Adoption, Implementation, and Maintenance, following the RE-AIM

framework [38], and Guatemala is also capturing contextual factors using the Practical, Robust, Implementation and Sustainability Model (PRISM) [41]. Both Guatemala and Kenya are collecting data on acceptability, appropriateness, and feasibility of the program and both are doing cost-effectiveness analyses. The Kenya team is using the Saunders Framework to guide its implementation assessment, which includes metrics of the referral process as well as a process evaluation [42]. Vietnam is using the Promoting Action on Research Implementation in Health Services model (PARiHS) to guide implementation and evaluation of their multi-level intervention within a de-centralized health system with semi-autonomous clinical delivery systems [43, 44]. Nepal plans to document program reach in terms of collaborations with international and regional partners, consolidation of members in different teams, and policy briefs for addressing the gap in health assessment.

In addition, all Hy-TREC sites and Malawi collected clinical outcomes to evaluate program effects. These included biomedical outcomes (blood pressure means, percent with controlled or uncontrolled blood pressure, change in CVD risk, mortality, hospital admissions, CVD complications, CVD risk factor prevalence, medication adherence, markers of adiposity) and behavioral outcomes (lifestyle behaviors such as diet, physical activity, alcohol or tobacco use, quality of life, and stage of change).

## Needs assessment

All projects conducted a needs assessment. TREIN sites focused needs assessments on general requirements for country-specific hypertension interventions, with a focus on documenting gaps in infrastructure, staffing, and or training. For example, Rwanda used the Context Assessment for Community Health (COACH) questionnaire with healthcare providers to design context-specific training [45]. At Hy-TREC sites, needs assessments were targeted towards obtaining information key to study or program implementation. In Ghana, a practice capacity survey, guided by CFIR [36], was conducted to assess health worker and facility readiness for study participation, and the Vietnam team conducted key informant interviews and focus group discussions with community health workers, program coordinators, local health managers, and hypertension patients to identify gaps in care and intervention targets at the study sites. Similarly, Guatemala conducted a multilevel qualitative participatory needs assessment, applying the WHO health system building blocks framework; alongside the needs assessment the team carried out workshops with health authorities, providers, and patients to adapt the intervention to the Guatemalan context [16, 46]. Both India and Kenya employed a mixed methods approach for their needs assessments, with India collecting data through a community epidemiological survey and qualitative interviews and Kenya conducting observational process mapping, a gap assessment, baseline referral network analysis and qualitative interviews (community meetings, key informant interviews, focus group discussions). Guatemala and India's needs assessments both included facility assessments at planned intervention sites.

## Changes due to the COVID pandemic

All projects reported some adaptations to timelines due to the pandemic, and the most common shift was adapting trainings, data collection, and meetings from in-person to virtual formats (Table 3). Changes were made to allow continuation of the project while protecting the health of the study team, trainees, and partners. Sites reported an increase in phone/internet contacts, with some sites setting up regular calls to maintain project momentum. Later in the pandemic, as in-person meetings became feasible at some sites, the size of gatherings was decreased to enable social distancing and some activities were carried out in a hybrid format.

**Table 3. Adaptations to capacity building activities at TREIN and Hy-TREC sites due to COVID-19 pandemic.**

| | TREIN (Rwanda, Malawi, Nepal) | Hy-TREC (India, Kenya, Ghana, Guatemala, Viet Nam) |
|---|---|---|
| Tools: | • Most work done via phones and computers<br>• Updated network connectivity<br>• Internet continued to be provided to PIs and senior faculty which was key when working remotely<br>• Transport continued to be provided as needed | • Telephone call system added to monitor participants, for example [47]<br>• Increased calls to participants to inform them of delays<br>• Increased sanitation of equipment |
| Skills: Training | • Skills-focused workshops delivered remotely<br>• Community-awareness campaigns delivered via social media, radio, and TV<br>• Weekly calls to connect members of the team and discuss publications and training<br>• Maintained expert linkages—just switched to virtual<br>• Meetings and workshops shifted to Zoom during 1st and 2nd wave<br>• Trainings not affected, but shifted to Zoom or in-person (with COVID-19 prevention measures) when cases low<br>• Closing of clinics disrupted some trainee's research projects, but they resumed activities as soon as possible<br>• Most trainings conducted pre-COVID-19; training of fellows done remotely via U.S. site<br>• Planned training of fellows in the U.S. canceled and done remotely | • Training activities were moved to Zoom or small group in-person sessions resulting in more time and manpower needed to implement<br>• Training done in batches over a period of two weeks instead of three days<br>• Health managers training delayed because of COVID obligations<br>• Trainings modified to one day in person + remote support with phone or WhatsApp<br>• Small group and phone/digital activities continue<br>• Updated standard operating procedure with COVID-19 related changes and conducted virtual training of staff on updates and best practices |
| Staff and Infrastructure | • Mentor-mentee interactions virtual | • Change in PI because of COVID-related death<br>• Changes in mentor team due to COVID-related changes in duties<br>• MOH implementing staff shifted focus to COVID responsibilities, including testing and vaccination |
| Structures, Systems, and Roles | • Still able to recruit high level trainees and engage Ministry of Health<br>• Meetings with Ministry of Health and other partners done online via various virtual meeting platforms<br>• NHLBI in person site visits canceled<br>• Fellows presented work (posters and presentations) online | • Meetings canceled or shifted online<br>• Despite challenges integration with Government of India NCD portal complete<br>• Health ministry remains engaged<br>• Health area directors and Ministry of Health needed to carry out intervention |
| Metrics | • Qualitative data collection done via telephone instead of in-person<br>• Rapid assessment of infrastructure and readiness halted | • Midline surveys delayed. Deciding if in person or phone data collection will be done is a moving target<br>• 6-month follow-up most affected with considerable delays of 3 to 5 months<br>• Project on hold for one year with only training activities occurring. Field activities resumed April 2021 |
| Needs Assessments | • Closing of the College of Medicine and health centers shifted timeline of needs assessment-related activities.<br>• Some health facilities could not be visited to collect data. Reporting used data from before the shut-down | • Shifted needs assessment data collection (could not conduct year 3)<br>• Conducted pre-COVID-19, so no change |

Key: Hy-TREC = Hypertension Outcomes for T4 Research in Lower Middle-Income Countries; TREIN = T4 Translation Research Capacity Building Initiative in Low Middle-Income Countries; PI = Principal Investigator; NCD = Non-communicable Disease; NHLBI = National Heart, Lung, and Blood Institute, National Institutes of Health

## Discussion

In this manuscript we described the capacity building activities in the eight programs making up the GRIT Consortium. Across the consortium, individual projects had the largest number of specific activities targeting the more technical and relatively simpler upper levels of the Capacity Pyramid—Tools and Skills. Sites increased tools available for office work and communication, data analysis and management programs, educational materials, and medical care

and outcomes assessment (e.g., blood pressure monitors, weighing scales). Hy-TREC sites—locations with implementation research projects—reported using implementation science frameworks as tools to assess implementation outcomes, most commonly RE-AIM [38], but also PRECEDE-PROCEED [37], PRISM [41], and CFIR [36]. All sites included both formal and informal training to build skills, with formal training programs on hypertension planned at 75% of the sites, evaluation/data collection and research ethics at 50% of sites, and dissemination and implementation research at 25% of sites. Sites were also invited to enroll trainees, researchers, and others in a MOOC on dissemination and implementation research, further expanding the reach related to training on this topic. In addition, seven sites included informal training through seminars and conferences.

GRIT Consortium projects also built capacity in the bottom tiers of the Capacity Pyramid, those more focused on the harder to change, sociocultural and structural aspects of capacity building. These include Staff and Infrastructure and Structures, Systems, and Roles, the base of the pyramid. Activities focused on staffing included mentoring of early stage team members by more experienced investigators, both from the in-country and U.S. project investigators (at 75% of sites) and providing protected time for investigators through the project grant funds (7/8 sites). Within the Structures, Systems, and Roles category, studies leveraged existing staff and trainees to assist in capacity building activities, created additional research fellow positions, and collaborated with the Ministry of Health and universities to implement capacity building activities and implementation research projects. All sites planned to disseminate their project findings through local media, conferences, publications, and key stakeholders (e.g., community members, health workers, and public health officials).

In a review of capacity building activities to improve implementation of evidence-based interventions in HICs, the authors reported that most studies focused on providing technical assistance and tools (e.g., manuals, evaluation tools) targeted to the specific intervention being delivered [48]. The factors that most influenced effectiveness of the capacity building strategies included setting-level factors such as resources and willingness to implement the provided program [48]. Although that review focused on research in HICs, these factors are also likely to impact work in LMIC settings. Lack of financial, personnel, and material resources are one of the strongest barriers to building research capacity in LMICs [10, 32]. Across the GRIT network, studies used funds to overcome some resource-related barriers, purchasing necessary tools, computers and software, and other needed equipment, as well as providing protected research time for LMIC investigators. The Hy-TREC studies, which focused on delivering evidence-based interventions in community settings, leveraged strong international partnerships to ensure interventions were designed and implemented in ways that were culturally and setting appropriate, acceptable within the communities of practice, and could be sustained and disseminated. Follow-up is needed to assess if these efforts will result in long-term improvements in implementation research capacity and post-intervention adoption of these interventions (if proven successful) in clinics or communities.

By contrast, a systematic review and meta-analysis of programs seeking to institutionalize research capacity in LMICs, described a larger variety of activities across studies to reach these goals [49]. This includes activities not widely used in GRIT projects, such as formalized education programs (masters degrees or diplomas), formalized programs for research-oriented skills building (grant or manuscript writing, see also [8]), research support systems for grant or data management, and analysis not linked to specific implementation research studies [49]. However, other capacity building activities were similar across the GRIT network and the studies included in the systematic review [49], including non-credit courses, online education, offering investigators protected time, setting up research groups and collaborative networks, and facilitating both in-country and international research collaborations. Also, although GRIT

Consortium projects did not provide formalized training on research skills, research mentors worked with trainees to build capacity in conducting, reporting, and presenting implementation research.

Even though the GRIT Consortium capacity building activities are comprehensive, additional activities are needed to foster the creation and maintenance of sustainable, relevant, and maximally effective implementation research infrastructure. In HICs, mentorship is often provided by a team of individuals, each serving different roles in helping the mentee reach her/his career, research, and personal goals. By contrast, in LMICs, the number of skilled mentors is typically limited, placing a large burden on existing mentors to provide mentorship and support for all their mentees' needs [50]. To do this effectively, however, these mentors need more than technical knowledge and research skills (e.g., conducting research, manuscript writing, grant writing); they require the leadership skills to succeed and in turn, to help their mentees grow and thrive [10, 32, 51]. To be an effective research leader, individuals require training in a multitude of areas including goal setting, effective communication, skill assessment and building, and fostering diversity, integrity, ethical behavior and research, and managing change [50], as well as the ability to successfully train others in these areas.

Building leadership skills and competencies while building technical skills will also empower mentors to champion the needs of the research community at their institutions to provide necessary resources including financial support (including protected research time), research infrastructure and staffing, access to continued training and necessary oversight [10, 32, 52, 53]. Even the most effective and well-trained research mentor is limited by the organization in which she or he works. A culture of research support, including policy supporting implementation of proven interventions and continued training to build and maintain research capacity, should be prioritized by universities, research institutions, Departments of Health, and local, regional, and national governments [54, 55]. The GRIT Consortium studies all included key stakeholders and public health leaders as partners or steering committee members, reflecting the understanding of leveraging these partnerships and working with policy makers. However, like other capacity building activities in LMICs [49], the GRIT network of programs and studies relied on funding from a HIC. Although providing funding in low-resource settings is often necessary to kickstart capacity building efforts, there is little evidence on long-term sustainability of these arrangements. Continued evaluation of the impacts of the GRIT Consortium projects will be vital to understanding the long term outcomes and impact of the projects and initial funding scheme.

There was variability in the capacity building activities across GRIT Consortium sites, with activities planned to be responsive to contextual factors and the needs of each country setting. All GRIT projects included a needs assessment to guide the program/study design and identify gaps in research/implementation science capacity. Methods differed across projects, but all incorporated feedback from a variety of stakeholders and assessed needs at multiple levels of the research, healthcare, and/or public health system. All Hy-TREC sites and one TREIN site included biomedical and behavioral outcome evaluation to describe the effectiveness of the implementation study or training outcomes. Most sites planned to evaluate program implementation using widely known and applied frameworks (e.g., RE-AIM [38], Saunders Framework [42]) and a few sites planned cost-effectiveness and acceptability assessments. In addition, all sites included implementation outcomes focused on assessing training and capacity building activities (i.e., number of trainees, evaluation of leadership support and organizational capacity). Across the GRIT Consortium, programs reflect the current shift in capacity building [32] from that focusing primarily on individual training (e.g., through graduate or post-graduate programs) to a more holistic, systems-level approach seeking to improve capacity at individual, organizational, local, national, and international levels. This differs from

other published work showing that few capacity building interventions include rigorous approaches to designing or testing the applied strategies for capacity building [48, 49].

Although there were impacts on project timelines and disruptions to clinic-based data collection, GRIT projects were able to pivot and continue activities throughout the COVID-19 pandemic. Common reported adaptations were moving to online or phone-based platforms for meetings, participant management, and data collection, planning in-person gatherings to be responsive to current pandemic regulations and safety protocols, and providing team members with protective equipment when in-field activities were feasible. Shifts in timeline and inability to access clinics during acute COVID-19 waves did not prevent data collection, only delayed it. Projects reported that they were able to maintain involvement of the Ministry of Health despite impacts of the pandemic. Some projects were forced to make changes to the investigative teams because of increased healthcare-related responsibilities of physician scientists on the team and in one case, the death of a study's contact PI. Overall, COVID-related adaptations were similar across GRIT projects, with observed differences based on variation in planned program or intervention activities and not location-specific barriers. These modifications allowed the project teams to continue their work and reach study goals, while minimizing risk of COVID transmission and adhering to local COVID-19 guidelines. Other organizations focused on capacity building in LMICs also describe similar project impacts and program changes to maintain capacity building work in the face of the COVID-19 pandemic [56, 57]. The pandemic has brought needed attention to inequities globally [58–61], and the need for programs to build capacity, improve equity, and promote equitable partnerships, locally, nationally, and internationally is clearer than ever. The GRIT Consortium projects, led, by in-country PIs, with a focus on describing and addressing locally identified gaps in implementation research capacity, may be a good model of how to start building better global partnerships.

The research presented in this article has several strengths. It describes the capacity building activities across a global network of studies and training programs focused on building implementation research capacity, strengthening global partnerships, and providing cross-country training and mentorship in global implementation research to early stage investigators. The authors describe these activities using an established framework for systemic capacity building [33]. Importantly, this adds to the limited reporting on capacity-building activities for implementation research in LMICs focused on chronic diseases, in this case hypertension. The authors also collected data describing project adaptations due to the COVID-19 pandemic. There are two potential limitations to this analysis. Although the authors are unable to report on the evaluations of these capacity building efforts or adaptations to activities during the COVID-19 pandemic in this manuscript, that data is forthcoming from the sites and will be shared in the future. Also, data is self-reported by sites using open ended survey items with only limited guidance on which activities to include in each category. Although the authors tried to actively engage site representatives in data reporting, data from the sites could be missing important nuance. Future studies of this type should consider mixed methods data collection to explore capacity building activities and their impacts.

## Conclusion

The GRIT Consortium included projects in LMICs across the globe which leveraged strong LMIC-HIC partnerships and baseline needs assessments to design capacity building activities and interventions that were responsive to the needs of the community and current evidence for hypertension management. Although long-term evaluation is needed to understand the impacts of these programs on research capacity, collaborations, and program longevity, they provide a model for planning capacity building activities in low-resource settings.

## Supporting information

**S1 Appendix. Capacity building activity reporting tool.**
(DOCX)

**S1 Data. Data reports.**
(PDF)

**S1 File. PLOS questionnaire on inclusivity in global research.**
(DOCX)

## Acknowledgments

The authors would like to acknowledge the tremendous and impactful contributions that Dr. Jacob Plange-Rhule made to capacity-building efforts and the hypertension program in Ghana and within the GRIT Consortium.

## Author Contributions

**Conceptualization:** Mary Beth Weber, Ana A. Baumann, Ashlin Rakhra, Constantine Akwanalo, Kezia Gladys Amaning Adjei, Josephine Andesia, Kingsley Apusiga, Duc A. Ha, Mina C. Hosseinipour, Adamson S. Muula, Hoa L. Nguyen, LeShawndra N. Price, Manuel Ramirez-Zea, Annette L. Fitzpatrick, Meredith P. Fort.

**Data curation:** Mary Beth Weber, Ana A. Baumann, Ashlin Rakhra, Constantine Akwanalo, Kezia Gladys Amaning Adjei, Josephine Andesia, Kingsley Apusiga, Duc A. Ha, Mina C. Hosseinipour, Adamson S. Muula, Hoa L. Nguyen, Manuel Ramirez-Zea, Annette L. Fitzpatrick, Meredith P. Fort.

**Formal analysis:** Mary Beth Weber, Ana A. Baumann, Ashlin Rakhra, Annette L. Fitzpatrick, Meredith P. Fort.

**Investigation:** Ana A. Baumann, Ashlin Rakhra.

**Methodology:** Mary Beth Weber, Hoa L. Nguyen, LeShawndra N. Price, Manuel Ramirez-Zea, Annette L. Fitzpatrick, Meredith P. Fort.

**Project administration:** Ashlin Rakhra.

**Validation:** Ashlin Rakhra.

**Visualization:** Ana A. Baumann, Ashlin Rakhra.

**Writing – original draft:** Mary Beth Weber.

**Writing – review & editing:** Mary Beth Weber, Ana A. Baumann, Ashlin Rakhra, Constantine Akwanalo, Kezia Gladys Amaning Adjei, Josephine Andesia, Kingsley Apusiga, Duc A. Ha, Mina C. Hosseinipour, Adamson S. Muula, Hoa L. Nguyen, LeShawndra N. Price, Manuel Ramirez-Zea, Annette L. Fitzpatrick, Meredith P. Fort.

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
