## [Decision Letter · Decision Letter 0]

22 May 2023

PGPH-D-23-00461

Global implementation research capacity building to address cardiovascular disease: an assessment of efforts in eight countries

Dear Dr. Weber

Thank you for submitting your manuscript to PLOS Global Public Health. After careful consideration, we feel that it has merit but does not fully meet PLOS Global Public Health’s publication criteria as it currently stands. Therefore, we invite you to submit a revised version of the manuscript that addresses the points raised during the review process.

Please ensure that all the comments given by the two reviewers are addressed sufficiently. Particularly, as both reviewers pointed out below, please revise the discussion section of the manuscript to illustrate why the experiences are important, and how they can contribute to the current knowledge in addressing cardiovascular disease in low and middle-income countries. Please critically edit the manuscript, including the tables, to fully meet PLOS Global Public Health’s publication criteria.

Please submit your revised manuscript by 03 July 2023 11:59 PM.  If you will need more time than this to complete your revisions, please reply to this message or contact the journal office at globalpubhealth@plos.org. Please include the following items when submitting your revised manuscript:

We look forward to receiving your revised manuscript.

Kind regards,

Azeb Gebresilassie Tesema, Ph.D.

Academic Editor

Journal Requirements:

1. Please include a complete copy of PLOS’ questionnaire on inclusivity in global research in your revised manuscript. Our policy for research in this area aims to improve transparency in the reporting of research performed outside of researchers’ own country or community. The policy applies to researchers who have travelled to a different country to conduct research, research with Indigenous populations or their lands, and research on cultural artefacts. The questionnaire can also be requested at the journal’s discretion for any other submissions, even if these conditions are not met.  Please find more information on the policy and a link to download a blank copy of the questionnaire here: https://journals.plos.org/plosone/s/best-practices-in-research-reporting. Please upload a completed version of your questionnaire as Supporting Information when you resubmit your manuscript.

2. Please provide separate figure files in .tif or .eps format only and remove any figures embedded in your manuscript file. Please also ensure all files are under our size limit of 10MB.

Additional Editor Comments (if provided):

Reviewers' comments:

**Comments to the Author**

1. Does this manuscript meet PLOS Global Public Health’s publication criteria? Is the manuscript technically sound, and do the data support the conclusions? The manuscript must describe methodologically and ethically rigorous research with conclusions that are appropriately drawn based on the data presented.

Reviewer #1: No

Reviewer #2: Yes

2. Has the statistical analysis been performed appropriately and rigorously?

Reviewer #1: N/A

Reviewer #2: N/A

3. Have the authors made all data underlying the findings in their manuscript fully available (please refer to the Data Availability Statement at the start of the manuscript PDF file)?

Reviewer #1: Yes

Reviewer #2: Yes

4. Is the manuscript presented in an intelligible fashion and written in standard English?

Reviewer #1: Yes

Reviewer #2: Yes

5. Review Comments to the Author

Reviewer #1: The study tackles an important research topic capturing data from multiple countries. However, it requires substantial work to make it suitable for publication. I would like to highlight the following general and specific comments to the authors.

General/major comments

1. The research question for the study is not adequately substantiated. The introduction section describes the benefit of the study as a learning experience for researchers, public health practitioners and clinicians. However, the manuscript did not identify this as a knowledge gap that warrants further research.

2. The study provides detail description of project activities from the GRIT sites. However these represent superficial description of the phenomenon lacking sufficient depth and technical rigour.

3. The study employed Potter and Brough model for analysis of the data. However, detail description of the analysis process is lacking.

4. The discussion section repeats the results from the study. Analysis of the results against previous work and the contribution of the study to fill or modify current knowledge in the field is not adequately described.

Specific

1. There are some typos grammatical errors that require correction (e.g. line 103, 'tier', not 'tiers'; line 141 'includes', not 'including'

2. Abbreviations need to be written in full when introduced for the first time (e.g. line 149 'IS process...')

3. The study used data from different studies conducted in the sites. It is not clear if ethics approval of these individual studies offered use of the data for secondary analysis such as for this study.

4. The study mentions the activities as easier and harder to implement. What is the criteria to categorise project activities as easier or harder to implement? Similarly 'formal' and 'informal' training requires definition.

5. In table 2, do empty cells meant the activity is not performed in the corresponding site? An X and tick symbols could provide better readability.

6. The study includes some subjective conclusions such as 'good model' or 'proven' frameworks. What measures did the authors use to report the GRIT model as a good model? What criteria was used to describe the frameworks as proven frameworks?

7. While the projects have made adaptations following COVID, it is not clear what the impact of such adaptations were on the outcomes of capacity building activities.

8. Some of the arguments described in the discussion section lack citation (e.g. the first sentence in paragraph two, page 20; the sentence in line 395 page 22; the sentence in line 402 page 22)

9. Evaluation of the capacity building activities was not conducted and this has been indicated as a limitation. However, the method section describes evaluation of the activities.

Reviewer #2: This is an interesting study to describe the capacities building activities for hypertension management in LMICs, and the changes and adaptions due to COVID-19 pandemic. My comments:

1. It will be great to have some background descriptions on the social, economic and political factors that influence the hypertension management in the countries, which will be helpful to understand the context.

2. In Table 3, better to specify the adaptions to capacity building activities by each country, since each country might have different scenario.

3. In Discussion, it will be better to include suggestions to improve capacity building skills in different countries. This will provide insights for further program conduction in these countries.

6. PLOS authors have the option to publish the peer review history of their article (what does this mean?). If published, this will include your full peer review and any attached files.

**Do you want your identity to be public for this peer review?** For information about this choice, including consent withdrawal, please see our Privacy Policy.

Reviewer #1: No

Reviewer #2: No

---

## [Editor Report · Decision Letter 1]

3 Aug 2023

Global implementation research capacity building to address cardiovascular disease: an assessment of efforts in eight countries<o:p></o:p>

PGPH-D-23-00461R1<o:p></o:p>

Dear Dr. Weber <o:p></o:p>

We are pleased to inform you that your manuscript 'Global implementation research capacity building to address cardiovascular disease: an assessment of efforts in eight countries' has been provisionally accepted for publication in PLOS Global Public Health.<o:p></o:p>

Before your manuscript can be formally accepted you will need to complete some formatting changes, which you will receive in a follow up email. A member of our team will be in touch with a set of requests.<o:p></o:p>

Please note that your manuscript will not be scheduled for publication until you have made the required changes, so a swift response is appreciated.<o:p></o:p>

IMPORTANT: The editorial review process is now complete. PLOS will only permit corrections to spelling, formatting or significant scientific errors from this point onwards. Requests for major changes, or any which affect the scientific understanding of your work, will cause delays to the publication date of your manuscript.<o:p></o:p>

If your institution or institutions have a press office, please notify them about your upcoming paper to help maximize its impact. If they'll be preparing press materials, please inform our press team as soon as possible -- no later than 48 hours after receiving the formal acceptance. Your manuscript will remain under strict press embargo until 2 pm Eastern Time on the date of publication. For more information, please contact globalpubhealth@plos.org.<o:p></o:p>

Thank you again for supporting Open Access publishing; we are looking forward to publishing your work in PLOS Global Public Health.<o:p></o:p>

Best regards,<o:p></o:p>

Azeb Gebresilassie Tesema, Ph.D.

Academic Editor

PLOS Global Public Health<o:p></o:p>